**Subject Category:**
Biology (whole organism)

ecology

camera trap, wildlife monitoring, ecological survey methods, meta-analysis

**Author for correspondence:**
Oliver R. Wearn
e-mail: oliver.wearn@ioz.ac.uk

# Snap happy: camera traps are an effective sampling tool when compared with alternative methods

Oliver R. Wearn[1] and Paul Glover-Kapfer[2]

[1]Institute of Zoology, Zoological Society of London, Regent's Park, London, UK
[2]WWF-UK, The Living Planet Centre, Rufford House, Brewery Road, Woking, UK

ORW, 0000-0001-8258-3534

Camera traps have become a ubiquitous tool in ecology and conservation. They are routinely deployed in wildlife survey and monitoring work, and are being advocated as a tool for planetary-scale biodiversity monitoring. The camera trap's widespread adoption is predicated on the assumption of its effectiveness, but the evidence base for this is lacking. Using 104 past studies, we recorded the qualitative overall recommendations made by study authors (for or against camera traps, or ambiguous), together with quantitative data on the effectiveness of camera traps (e.g. number of species detected or detection probabilities) relative to 22 other methods. Most studies recommended the use of camera traps overall and they were 39% more effective based on the quantitative data. They were significantly more effective compared with live traps (88%) and were otherwise comparable in effectiveness to other methods. Camera traps were significantly more effective than other methods at detecting a large number of species (31% more) and for generating detections of species (91% more). This makes camera traps particularly suitable for broad-spectrum biodiversity surveys. Film camera traps were found to be far less effective than digital models, which has led to an increase in camera trap effectiveness over time. There was also evidence from the authors that the use of attractants with camera traps reduced their effectiveness (counter to their intended effect), while the quantitative data indicated that camera traps were more effective in closed than open habitats. Camera traps are a highly effective wildlife survey tool and their performance will only improve with future technological advances. The images they produce also have a range of other benefits, for example as digital voucher specimens and as visual aids for outreach. The evidence-base supports the increasing use of camera traps and underlines their suitability for meeting the challenges of global-scale biodiversity monitoring.

# 1. Introduction

Camera traps have come a long way from their beginnings in wildlife photography more than 100 years ago and are now a ubiquitous tool in ecology and conservation, with several hundred scientific studies now published each year using them [1]. The last decade, in particular, has seen the camera trap move from being a niche tool primarily for monitoring big cats (e.g. [2–4]), to taking centre stage in broad-spectrum surveys of whole communities of mammals (e.g. [5–7]). The camera trap has now been adopted by a number of large-scale biodiversity monitoring programmes (e.g. [8,9]) and underpins the Wildlife Picture Index [10], which informs progress towards the Convention on Biological Diversity's Aichi Target 12 (https://www.bipindicators.net/indicators/wildlife-picture-index).

The camera trap's widespread adoption is predicated on an assumption of its effectiveness. Certainly, a number of methodological case studies have found them to compare favourably with other methods for surveying species richness [11–13], species distributions [14–16], relative abundance [17–19] and animal density [20–22]. The key strength of the modern camera trap is its ability to remain in the field for protracted periods of times (on the order of months at a time, if needed), continuously registering detections day and night, with relatively little input from human operators required. This means that even the rarest events, such as nest predation or the passage of an apex predator through an area, can be quantified and studied. Camera traps also facilitate this without significant disturbance to study animals, except for the emission of sound and light [23].

However, a comprehensive assessment of the effectiveness of camera traps compared with alternative methods has never been done. For example, a number of studies have found that camera traps can sometimes be outperformed by other survey methods in certain contexts (e.g. [24–26]). Camera traps are certainly no panacea for surveying wildlife and suffer from a number of limitations in their current form [27], most importantly their relatively high initial cost (typically \$200–500 per unit) and the relatively small area that they monitor (notional detection zones of camera traps are typically less than $2 \times 10^{-4}$ km$^2$). A critical appraisal of camera traps, as has already been done for other technologies (e.g. GPS telemetry: [28]), is now long overdue.

We here draw upon the substantial body of evidence that has accumulated on the effectiveness of camera traps to ask: (i) which sampling methods do camera traps compare most favourably with, and (ii) can we identify the specific study contexts in which camera traps are most effective relative to other methods? In particular, we expected camera trap effectiveness to depend strongly on study objectives, but we also tested specific hypotheses related to the equipment and protocol used, the species under study, and the habitat and location of the study. Ours is the first global assessment of camera trap effectiveness to date, and offers broad insights into when and where this technology can be most effectively deployed.

# 2. Material and methods

## 2.1. Literature search

We searched the scientific literature for studies which have compared camera traps with other ecological survey methods, focusing on peer-reviewed journal articles. To do this, on 29 January 2018, we searched the Core Collection of the Web of Science (http://webofknowledge.com) with the query 'camera trap*' OR 'camera-trap*' OR 'game camera*' OR 'trail camera*' OR 'scouting camera*' OR ('remote camera*' AND 'wildlife') OR ('automatic camera*' AND 'wildlife') OR ('automatic photograph*' AND 'wildlife') for the period 1969–2017. This generated a list of titles and abstracts ($n = 1981$) which we scanned for suitable studies.

We defined suitable studies as those which presented quantitative data comparing camera traps to one or more other survey methods and which drew explicit conclusions about their relative effectiveness. This included purely methodological studies, as well as ecological studies which also offered explicit comparisons of multiple survey methods as a secondary component of the paper. We only included studies using triggered camera traps, as opposed to time-lapse cameras. Where we were not sure if a study was suitable based on the title and abstract, we downloaded the full text to check the main body of text. We also followed the literature trail; if studies we read in turn cited other studies of potential interest, we assessed their suitability for inclusion as well.

This process yielded 100 peer-reviewed studies, as well as four unpublished technical reports that were cited by other studies, from which we attempted to extract data.

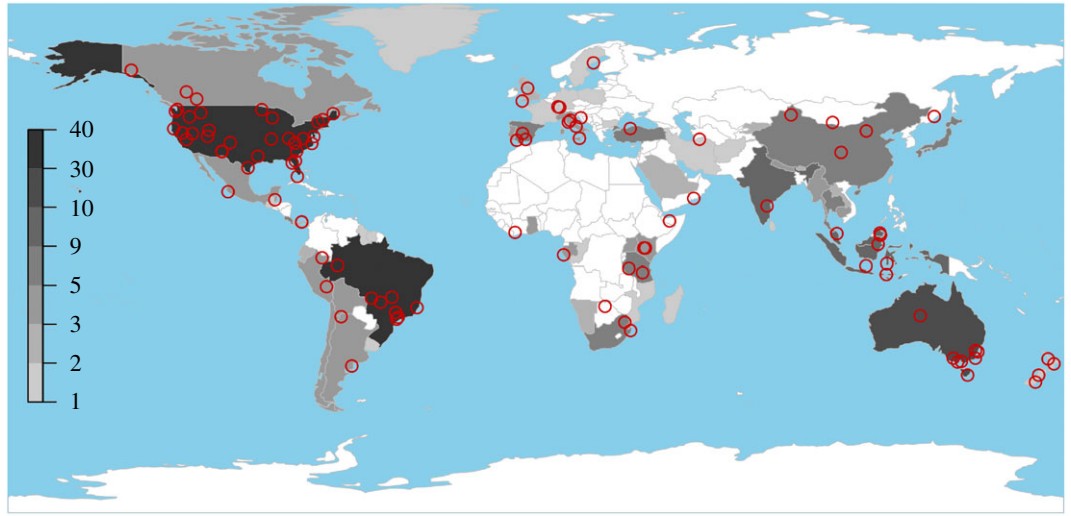

**Figure 1.** Locations of primary studies used in this study. The base map displays a measure of overall camera-trapping publishing output (the number of camera trap studies per country published between 2008 and 2014, from [47]).

## 2.2. Data extraction

For each of the 104 studies (figure 1) we identified from the literature search, we attempted to extract both qualitative and quantitative data on the effectiveness of camera traps relative to other survey methods. First, we made an overall qualitative assessment of whether the authors of the study had concluded that camera traps were the superior method for achieving the aims of the study. Author conclusions have previously been used to distil the findings of complex studies involving many variables (e.g. [29]). Although this is a coarse measure of camera trap effectiveness, subject to the attitudes and value systems of the authors of each study, we hypothesized that it would capture information not present in the quantitative data presented in each paper, in particular better capturing the authors' practical experiences of implementing each method.

To qualitatively assess author recommendations, we read each paper, focusing on concluding remarks in the Abstract and Discussion sections. We were able to categorize the author conclusions in most studies (61%) as either 'In support of camera traps' or 'In support of another method' on this basis. If camera traps were equal in performance to another survey method, then we scored the study as 'Ambiguous' (39% of studies). This category was also used if the authors presented heavily caveated conclusions (typically, concluding that one or other survey method was best in specific circumstances, such as particular weather conditions, or for specific species). Given the subjectivity inherent to this scoring, one of us (O.R.W.) did all the scoring in a consistent manner.

In addition to scoring the qualitative author conclusions, we also extracted quantitative data on the effectiveness of camera traps relative to other survey methods, which we fed into a meta-analysis. Suitable quantitative data included comparisons of the following metrics: (i) the number of species detected; (ii) detection rates (per sampling occasion, i.e. per trap night or as defined in the study); (iii) latencies to detection (the sampling effort required to register the first detection); (iv) detection probabilities (species or individual detection probabilities, estimated from occupancy or capture–recapture models, respectively); (v) the number of individuals detected (only possible in cases where individuals could be identified); (vi) the accuracy of state variable estimates (measured in percentage bias from a benchmark, such as a true abundance count); (vii) the precision of state variable estimates (measured using the coefficient of variation); (viii) implementation efforts (measured in person hours of labour), and (ix) costs. For costs, we took account of as many costs as the authors provided information for, typically including equipment, labour and associated field costs, but sometimes also including training, data entry and equipment repair costs. Costs were standardized in different ways across studies, in most cases per survey (e.g. the cost of obtaining a single-session abundance estimate) but in some cases per unit area sampled or per animal detection. If a study presented costs separately for successive sessions of sampling (e.g. costs for years 1–5), then we took the costs for the first session.

Survey methods being compared with camera traps were grouped into broad classes (see electronic supplementary material, table S1.1 in appendix S1 for definitions). Each study used a different experimental design to compare methods, depending on the specific objectives and methods involved,

but most studies attempted to spatially and temporally match the deployment of each survey method (e.g. co-deploying methods at the same sampling points). Where this was obviously not the case, we limited comparisons to metrics that we considered would still be valid. For example, if the temporal extents differed, we only considered metrics that are invariant to sampling effort (detection rate, latency to detection, detection probabilities and percentage bias).

We also extracted the sample sizes and variances associated with each data point, where these were presented or could be calculated from available information. However, this was only possible in a minority of cases (39% of data points). Where data were presented in figures, rather than in the main text or tables, we extracted the values using the open source tool 'Engauge Digitizer' (http://markummitchell.github.io/engauge-digitizer).

## 2.3. Quantifying relative camera trap effectiveness

For each pair of data points—one for camera-trapping and one for the other survey method—we calculated an effect size to capture the magnitude of the difference in performance between methods. We used the log response ratio [30,31] as our measure of effect size, which is appropriate when combining information across values measured on very different scales. It also has the practical benefit in our case of not requiring sample sizes or variances, which were unavailable in most cases.

We calculated the log response ratio as: $\ln(\text{Metric}_{\text{camera trap}}/\text{Metric}_{\text{other sampling method}})$, using the values we extracted in one of the nine categories of metric outlined above. For metrics in which lower values indicate better performance (latency to detection, percentage bias, coefficient of variation, implementation effort and cost), we instead took $\text{Metric}_{\text{camera trap}}$ as the denominator. Larger effect size values in this paper therefore always indicate that camera traps outperformed the survey method they were compared with. Log response ratios cannot be calculated when either the numerator or denominator is zero. This issue was mostly confined to the comparison of detection rates, and in practice means that the comparison is of detection rates when both methods detected the given species at least once. In total, we obtained 662 quantitative comparisons across 97 of the 104 studies in our sample.

To obtain the median effect sizes for each type of metric and for each survey method comparison, including estimates of uncertainty, we used a stratified bootstrapping approach (e.g. [32]). This approach deals effectively with the lack of independence between multiple comparisons provided by a single study, because in each bootstrap, only a single comparison per study (selected at random) is chosen. Bootstrap sampling with replacement ($n = 10\,000$) was done separately for each metric and each method with more than 15 comparisons available in the data. We were also *a priori* interested in effect sizes according to three other factors—the type of camera trap used (*film* or *digital*), whether baits or lures were used or not with the camera traps (*attractant* or *no attractant*), and the openness of the habitat (*closed* forested habitats or *open* shrubland, grassland and desert habitats)—and calculated bootstrapped median effect sizes in the same way as for the metrics and survey methods. We calculated 95% confidence intervals for the bootstrapped effect sizes from the quantiles of the distribution of bootstraps and judged effect sizes to be significant if the intervals did not overlap zero. For comparing across effect size estimates, we used *post hoc* pairwise Wald tests [33] and judged significant differences using an error rate ($\alpha$) of 0.05.

We could not weight the effect size for each comparison by its precision because the required data for this were not presented in most studies. This means that the precision of our overall effect size estimates may be reduced, but does not necessarily mean that the estimates are biased [34].

## 2.4. Hypothesis testing

We modelled the author recommendations and the effect sizes separately but tested a similar set of hypotheses. In particular, we hypothesized that studies which used digital instead of film camera traps, used an attractant with the camera traps and were done in closed instead of open habitats would be more effective. Habitat openness was thought to be important on the basis of anecdotal information that the most common type of camera trap sensor—the passive infrared (PIR) sensor—suffers from misfires and missed detections in open habitat, due to denser ground vegetation and higher temperatures, respectively [1]. We also hypothesized that studies which included small-sized animals (i.e. animals with a weaker infrared signal) would be less effective and modelled this using the minimum of the body weights of the focal species in a given study. The absolute latitude of the study area was also included to test if studies done in the tropics were less effective than those at higher latitudes, due to higher temperatures and possibly greater rates of equipment failure [1].

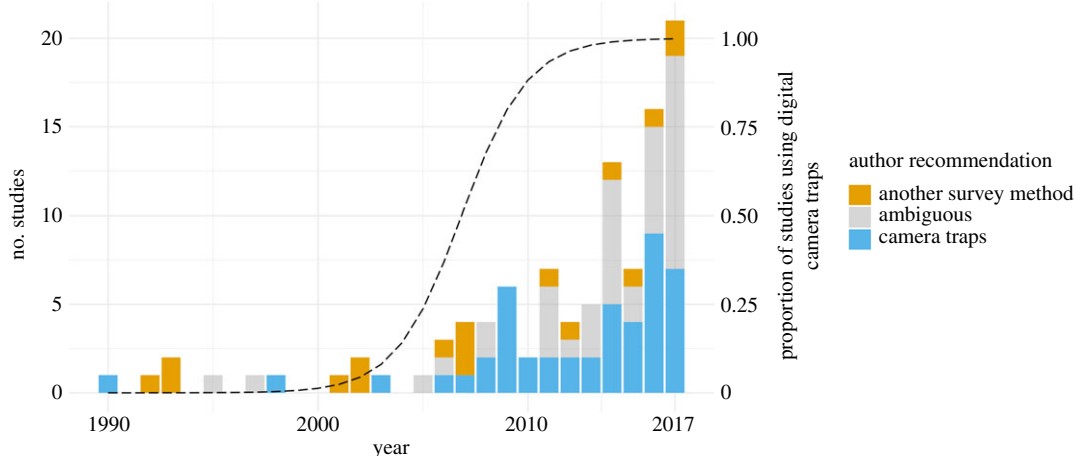

**Figure 2.** Author recommendations over time and the proportion of studies using digital camera traps over the same period. The proportion of studies using digital camera traps (dashed line) was predicted using a generalized linear model with a binomial response. This was based on the year fieldwork began, rather than the publication date, owing to the substantial lag between the two in some studies.

For the models of effect sizes, we also included covariates for the metric used and the comparison survey method used, but we could not include these covariates in the models of the author recommendations, because each study typically involved multiple metrics and/or methods. We tested for collinearity among the covariates using variance inflation factors, which provided no cause for concern (all factors less than 3; [35]).

We treated author recommendations as an ordinal response and used proportional odd models implemented in the *ordinal* package in R [36]. We fit the model *Recommendation ~ Camera type + Use of attractants + Habitat openness + Latitude + Minimum weight*, as well as all nested versions of this model (for a total of 32 models), using the MuMIn package in R [37]. The continuous covariates, *Latitude* and *Minimum weight*, were centred at their mean and scaled by twice their standard deviation [38]. We took the natural logarithm of the minimum body weights (before standardization), owing to the strong positive skew in these data. We did not consider interactions among variables, owing to a lack of data.

Important variables were those that appeared in the set of 'best' models based on information-theoretic criteria (i.e. models for which $\Delta AIC_c < 2$ [39]). The relative importance of each variable was assessed using the standardized parameter estimates [40] and by summing the Akaike weights of the models in which each variable appeared [39,41]. We calculated model-averaged parameter estimates (using 'natural averaging', to avoid shrinkage of estimates towards zero) and 95% confidence intervals, which incorporate model uncertainty as well as parameter uncertainty ([39,42], but see [43,44]).

We modelled effect sizes, i.e. log response ratios, using linear mixed-effects models in the *lme4* package in R [45]. The saturated model in this case was *ln(Response ratio) ~ Camera type + Use of attractants + Habitat openness + Latitude + Minimum weight + Metric + Comparison survey method*, as well as a random intercept for *Study* to account for the lack of independence among multiple measures made from the same study. As for the author recommendations, we fit all nested versions of this model (for a total of 128 models), assessed the relative importance of variables using the standardized parameter estimates and sums of Akaike weights, and calculated model-averaged parameter estimates and 95% confidence intervals.

We made all calculations and fit the models in R v. 3.4.1. [46].

## 3. Results

Studies took place in 32 countries, on all continents except Antarctica, with most studies occurring in the USA ($n = 33$), Brazil ($n = 9$) and Australia ($n = 8$). This reflects broader geographical trends in camera-trapping publishing output ([47]; figure 1). Studies spanned the period 1990–2017, with fieldwork conducted from the late 1980s to 2016. Across this period, film camera traps were gradually replaced by digital camera traps, with the latter becoming more commonly used from 2007 onwards (figure 2). Camera-trapping was compared with a wide range of other survey methods ($n = 22$ methods),

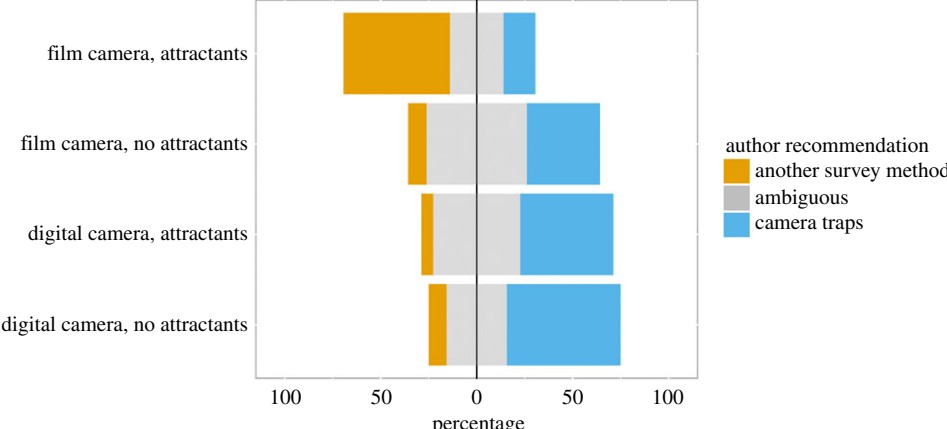

**Figure 3.** Author recommendations according to the type of camera used (film or digital) and whether attractants (i.e. baits or lures) were used or not. The percentage of studies has been calculated separately within each factor level combination.

**Table 1.** The set of 'best' models for the author recommendations and effect sizes (log response ratios), as identified using information-theoretic criteria ($\Delta AIC_c < 2$). Models of effect sizes all included a random effects term for *Study*.

| model | parameters | log-likelihood | $\Delta AIC_c$ | Akaike weight |
|---|---|---|---|---|
| author recommendations—proportional odds model | | | | |
| *Camera type + Use of attractants* | 4 | −98.42 | 0 | 0.30 |
| *Camera type + Use of attractants + Habitat openness* | 5 | −97.93 | 1.23 | 0.16 |
| *Camera type + Use of attractants + Latitude* | 5 | −98.31 | 1.99 | 0.11 |
| effect sizes—linear mixed-effects model | | | | |
| *Camera type + Habitat openness + Metric + Comparison survey method* | 17 | −1022.24 | 0 | 0.27 |
| *Camera type + Metric + Comparison survey method* | 16 | −1023.95 | 1.29 | 0.14 |

including methods as disparate as acoustic recording, eDNA, radio-tracking and local ecological knowledge (a full list of the methods is provided in electronic supplementary material, appendix S2).

## 3.1. Author recommendations

Most studies recommended the use of camera traps overall ($n = 46$), though many studies also found that other survey methods were equal in performance or presented caveated support for camera traps (i.e. 'Ambiguous'; $n = 41$). A minority of studies supported the use of other survey methods overall ($n = 17$). Three models were identified as the 'best' at explaining the variation in author recommendations (table 1). These contained the variables *Camera type* (importance = 0.99; model-averaged estimate of the difference between factor levels, with *digital* as the reference level = −1.42; 95% CI: −2.24 to −0.61), *Use of attractants* (importance = 0.84; estimate = −0.94; 95% CI: −1.74 to −0.14), *Habitat openness* (importance = 0.34; estimate = −0.40; 95% CI: −1.21 to 0.40) and *Latitude* (importance = 0.27; estimate = 0.20; 95% CI of the slope: −0.67 to 1.07). Digital camera traps were 4.1 times more likely to result in an affirmative author recommendation than film camera traps, in support of our hypothesis (figure 3). Studies which did not use attractants during camera-trapping were 2.5 times more likely to result in an affirmative author recommendation than those that did use attractants (figure 3), which is counter to our hypothesis that attractants would make camera traps more effective. This effect appears to have been driven, at least in part, by a cluster of studies on North American forest carnivores (e.g. [24,25,48]), which routinely baited their camera traps and found them to be less effective than other survey methods.

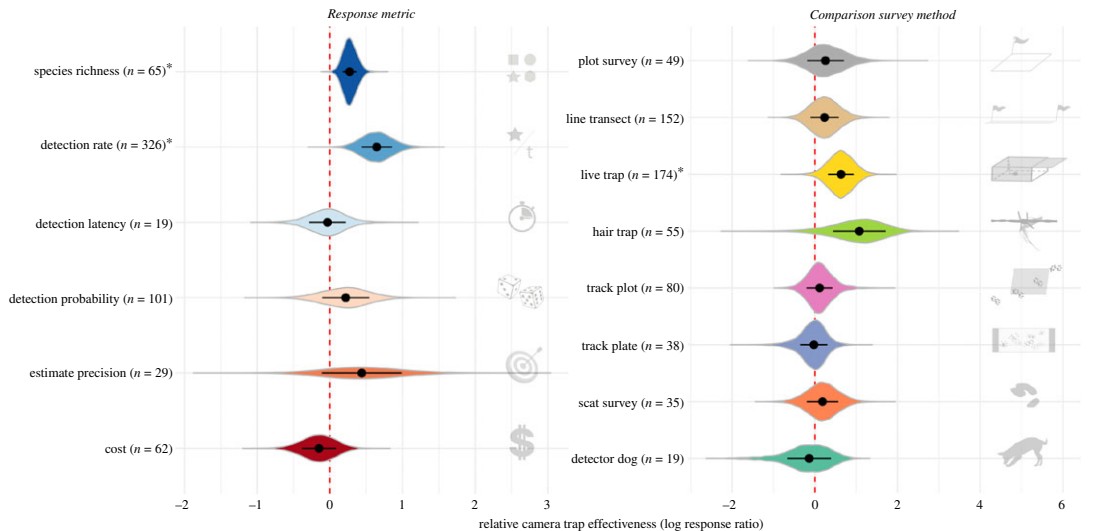

**Figure 4.** Bootstrapped effect sizes for each metric and survey method, with the distribution of bootstrap estimates shown in each case with a violin plot. Asterisks after the labels indicate cases in which camera traps were significantly more effective than the methods they were compared with. Points and error bars indicate medians and standard errors. Sample sizes are the number of effect sizes extracted from the primary literature, with non-independence among effect sizes from the same study accounted for with stratification during each bootstrap. The red vertical dashed line at zero indicates no difference in effectiveness.

## 3.2. Relative camera trap effectiveness

The bootstrapped effect size over the studies in the sample was 0.33, which corresponds to a 39% (95% CI: −2% to 100%) higher effectiveness of camera traps relative to the methods they were compared with. Digital camera traps, which have now completely superseded film-based models (figure 2), were 65% (95% CI: 2–169%) more effective than other methods. The effect sizes per metric and method showed that in no case were camera traps significantly inferior to other survey methods (figure 4; electronic supplementary material, tables S2.1 and S2.2 in appendix S2). On the contrary, camera traps were significantly more effective than other methods at detecting a higher number of species (31% higher effectiveness for *species richness*; 95% CI: 11–60%) and at quickly generating a large number of detections for individual species (91% higher effectiveness for *detection rate*; 95% CI: 29–192%). Similarly, camera traps were significantly more effective than live traps (88%; 95% CI: 1–239%).

In contrast with the author recommendations, the bootstrapped effect sizes calculated from the quantitative data in the same studies did not show a difference in camera trap effectiveness according to whether attractants were used ($z = 0.05$; $p = 0.96$; figure 5). Effect sizes were larger for studies using digital rather than film camera traps, and for studies in closed rather than open habitats, but the differences were not significant, given the large sampling variances (*Camera type*: $z = 1.45$; $p = 0.15$; *Habitat openness*: $z = 0.88$; $p = 0.38$; figure 5).

Mixed-effects modelling identified two models that were 'best' at explaining the variation in effect sizes (table 1). These models contained the variables *Metric* (importance = 0.930; see electronic supplementary material, table S2.3 in appendix S2 for estimates and 95% CIs), *Camera type* (importance = 0.928; model-averaged estimate of the difference between factor levels = −0.55; 95% CI: −0.91 to −0.18), *Comparison survey method* (importance = 0.89; see electronic supplementary material, table S2.4 in appendix S2 for estimates and 95% CIs) and *Habitat openness* (importance = 0.66; estimate = −0.41; 95% CI: −0.77 to −0.05). There was no support for the importance of *Use of attractants*, *Latitude* or *Minimum weight*. The lack of support for any effect of attractants again contrasts with the author recommendations.

# 4. Discussion

We found that studies comparing camera traps to other survey methods, in general, support the notion that camera traps are a highly effective wildlife survey tool. This was confirmed by the recommendations made by study authors as well as the quantitative data underlying the studies. The quantitative data showed that camera traps were effective across a range of different metrics, but were especially effective at detecting a wide range of species and at recording a large number of detections of focal species. This makes camera

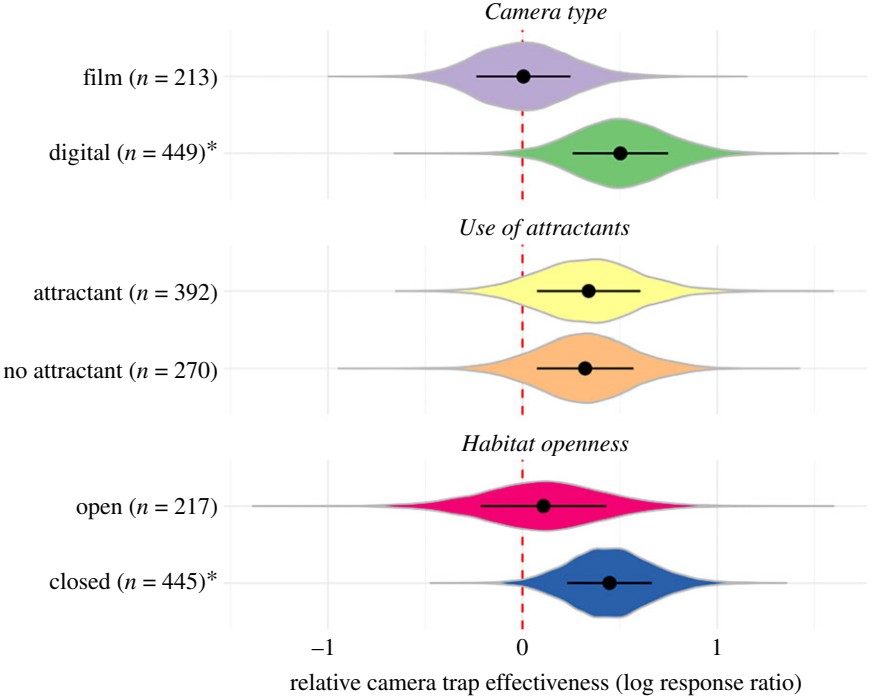

**Figure 5.** Bootstrapped effect size according to camera trap type, whether attractants were used and habitat openness. Asterisks after the labels indicate cases in which camera traps were significantly more effective than the methods they were compared with. Points and error bars indicate medians and standard errors. Sample sizes are the number of effect sizes extracted from the primary literature, with non-independence among effect sizes from the same study accounted for with stratification during each bootstrap. The red vertical dashed line at zero indicates no difference in effectiveness.

traps ideally suited for broad-spectrum biodiversity surveys of the kind often required for research or management purposes. Camera traps performed significantly better than live traps and were otherwise comparable in performance to other survey methods. Confirming a widely held belief, the transition to digital camera traps over the last 15 years has substantially increased the effectiveness of the method. There was also evidence that camera trap studies done in open habitats are less effective. Other characteristics of studies—the use of baits and lures, the body size of focal taxa and latitude—were not found to have a strong overall bearing on the effectiveness of camera traps, although this does not discount the fact they are likely to be important considerations in specific cases.

## 4.1. How do camera traps compare with other survey methods?

There was substantial variation in camera trap effectiveness depending on the method they were compared with, as evidenced by the bootstrapped effect sizes (figure 4) and the mixed-effects model results (table 1). Camera traps were most effective in comparison to live traps and hair traps, and least effective in comparison to detector dogs, although the difference was only significant in the case of live-trapping (see electronic supplementary material, appendix S3 for discussion of hair traps and detector dogs). Live traps performed poorly compared with camera traps in part because they are typically 'single-catch' traps, i.e. they require manual resetting after each capture, while camera traps are 'multi-catch' traps (e.g. [15,49,50]). Live traps also require an animal to be attracted by a bait or lure, interact with a foreign object in the environment (i.e. the trap) and be physically caught successfully. This sequence of events, which may fail for a wide range of reasons, contrasts with camera-trapping, which only requires an animal to enter the camera detection zone and trigger the sensor. As a result, live-trapping methods are typically optimized for a relatively narrow set of species (e.g. [50,51]). A major benefit of live-trapping, however, is that animals are physically caught, meaning that samples can be taken from animals, they can be marked for the purposes of studying population dynamics and they can be GPS- or radio-tagged.

## 4.2. Camera trap effectiveness depends strongly on study objectives

The most important predictor identified in the mixed-effects models was *Metric* (table 1), indicating that camera trap effectiveness varied widely depending on the type of measurement made (figure 4). Camera

traps were especially effective at detecting a large number of species, owing to the relatively low specificity of most camera trap sensors. The PIR sensors typical of commercial camera traps are known to be effective for any endothermic species larger than 100 g [1], and it is possible to detect even smaller species (including ectothermic species) at distances less than 1 m using specialized protocols [52–54]. Camera traps were also effective at making a large number of detections of focal species, owing to their capacity to record multiple detections without requiring maintenance. While this was less true of film camera traps, which were limited to a single roll of film, modern digital camera traps are now capable of recording more than 10 000 images on a single set of batteries. Camera traps were less effective than other methods at detecting species quickly (for example, in a rapid assessment survey) and cheaply, though in neither case was this difference significant (figure 4; see electronic supplementary material, appendix S3 for additional discussion).

## 4.3. Differences in camera trap effectiveness according to equipment and habitat

The type of camera trap used—film or digital—had the most important bearing on camera trap effectiveness (table 1), with film camera traps being much less effective (figures 3 and 5). In fact, after removing film camera traps from the sample, camera traps were overall significantly more effective than other methods. As well as the limited number of images that film camera traps could store, they were often paired with active infrared sensors, which have much smaller detection zones (they require animals to cross a beam of infrared) compared with PIR sensors, further limiting their effectiveness. Improvements in camera trap performance must be accounted for when making inferences about wildlife population changes over long time-frames from camera trap data (e.g. by accounting for detection probabilities). In addition, it suggests that next-generation camera trap technology (e.g. replacing PIR sensors with pixel-based change detection algorithms) might lead to similarly dramatic improvements in camera trap effectiveness [27].

Mixed-effects modelling of the effect sizes identified habitat openness as an important variable (table 1), with camera traps being more effective in closed habitats. Authors were not, however, more likely to recommend using camera traps in closed habitats. Most camera trap studies are done in closed habitats [27,55], perhaps because of their greater effectiveness in this habitat. We hypothesized that this is because PIR sensors are more effective in closed habitats, but it might also be because researchers have a more limited set of survey methods available to them in dense forest habitat (for example, aerial surveys of terrestrial animals are not possible) or because certain survey methods are less effective (e.g. line transects). Either way, it underlines the importance of applying best-practice methods to help address the potentially large number of problems that open environments can pose [1].

## 4.4. Limitations: experimental design and representativeness

During this review, we identified a number of shortcomings across studies. In particular, some studies suffered from poor spatial or temporal matching across the sampling methods being compared. We limited the impacts of this by comparing metrics that would be sensitive to this only across similar spatial or temporal extents. However, we did include studies which sampled over the same overlapping area, albeit not at exactly the same spatial point locations (this was necessary to compare sampling methods which are not point-based, such as line transects and plot surveys), and we also included studies which used the methods at different times (for example, in different months or different years). We anticipate that this will introduce additional noise into our data, but not undermine our main results.

We also included studies that did not standardize financial costs across the methods being compared (and instead we measured any cost differences as an effect size). This was because financial costs are one of several possible resources that might be a limiting factor for any given research team carrying out a camera trap survey, for example the time taken to complete a survey in the case of a rapid survey, or field labour in the case of a team which has limited personnel. Although many studies explicitly acknowledged the resource that they had matched across sampling methods (e.g. [12,49]), others did not, and in these cases the resources allocated to each sampling method were probably determined by accepted 'best-practice' (e.g. [19,22]), or by logistical factors external to the study. We only made comparisons across methods where it was sensible to do so, for example using a metric that was insensitive to resources (e.g. detection rate). However, it remains the case that our overall results are reflective of the resources allocated to each sampling method across the studies in our sample.

The set of studies in our sample reflects broader biases in where camera-trapping is done and what it is used for [27,47,55]. Therefore, studies from the USA, and on mammalian carnivores, are more common in our dataset than other types of study. If we are to make more general conclusions about the effectiveness of camera-trapping, irrespective of current geographical and taxonomic biases, then we would ideally have access to a much more balanced dataset. This will only be possible if steps are taken to address the broader geographical biases in conservation research [56].

## 4.5. An evidence-base for more effective wildlife surveys

More than a decade ago, Rowcliffe & Carbone [57] asked whether camera traps might have a bright future in ecological monitoring. We have established the evidence-base to show that they do. Across a range of metrics, camera traps are either comparable to or outperform existing alternative methods, and their performance is only likely to improve with future advances in camera trap technology (outlined in [27]). In addition, we have not considered here the considerable value that camera trap images and videos have as digital museum voucher specimens, and as powerful visual aids for outreach, science communication, lobbying and community engagement. The raw data recorded by other methods are typically much less rich in information (with the exception of DNA sequences) and usually do not provide information on group size, behaviour or other natural history details.

Camera traps have recently been advocated as a key component of a proposed global biodiversity monitoring network [58]. Camera traps are prime candidates because they allow for highly standardized data collection, they minimally disturb wildlife, and they generate data that can be up-scaled to regional or global scales while accounting for imperfect observation. Adding to this, we have here shown that they are also highly effective relative to other methods. We anticipate that the coming decade will be even brighter for camera-trapping than the last.

Data accessibility. All of the raw data used in this publication, including the author recommendations and calculated effect sizes, are available online in the Zenodo repository [59].
Authors' contributions. O.R.W. conceived and designed the study, analysed the data and led the manuscript writing. Both authors collected the data, edited the manuscript and gave final approval for publication.
Competing interests. We declare that we have no competing interests.
Funding. Primary funding for this study was provided by WWF-UK. O.R.W. was also supported by an AXA Research Fellowship during this work.
Acknowledgements. We thank all of the authors of the primary research we drew upon. We have included in the reference list those studies not cited in the main text [60–140]. We also thank Joanna Burgar and an anonymous reviewer for their constructive comments on the manuscript.

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
