## [Reviewer comments · Royal Society Open Science]

Review History

RSOS-181748.R0 (Original submission)

Review form: Reviewer 1 (Joanna Burgar)

Is the manuscript scientifically sound in its present form?

Yes

Are the interpretations and conclusions justified by the results?

Yes

Is the language acceptable?

Yes

Is it clear how to access all supporting data?

Yes

Do you have any ethical concerns with this paper?

No

Have you any concerns about statistical analyses in this paper?

Yes

Recommendation?

Accept with minor revision (please list in comments)

Comments to the Author(s)

See attached file (Appendix A).

Review form: Reviewer 2 (Tim Hofmeester)

Is the manuscript scientifically sound in its present form?

Yes

Are the interpretations and conclusions justified by the results?

Yes

Is the language acceptable?

Yes

Is it clear how to access all supporting data?

Yes

Do you have any ethical concerns with this paper?

No

Have you any concerns about statistical analyses in this paper?

No

Recommendation?

Accept with minor revision (please list in comments)

Comments to the Author(s)

I really enjoyed reading your manuscript entitled 'Snap happy: camera traps are an effective sampling tool compared to alternative methods'. I think it is a timely meta-analysis of the studies comparing camera trapping with other survey methods that confirms the big potential that is often attributed to camera traps. The methods for the meta-analysis are well described and appropriate statistics performed. I have only a few questions and minor suggestions for additions to the manuscript.

Material and methods

The authors made sure that only within study comparisons were made in order to reduce biases in the analyses due to differences in quantification of certain parameters (for example calculating detection probability per day or per week). This is further supported by the use of the log response ratio as the measure of effect size. If I am correct, this method assumes that parameters were calculated in the same way and on the same scale within studies. If the authors checked for this, it is good to make this explicit, otherwise, I think this needs checking to make sure that comparisons are made in a consistent way. [Lines 142-147]

Results

[Lines 223-225] You report a difference between factor levels, but which factor is taken as the baseline? I guess digital, but would be good to state.

For bait: where camera traps with bait compared to other methods with bait, and camera traps without bait with other methods without bait? Or could camera traps with bait be compared to other methods without bait and vice versa? I am wondering because of the counter intuitive results for baited cameras having a smaller difference with other surveys than un-baited cameras. Could this perhaps be because studies in which baited cameras were used were also more likely to use other methods that used bait, and the use of bait in general increased the effectiveness of any method, and therefore there is a smaller difference between methods? [Lines 228-234]

Discussion

[Lines 294-296] As it is a common misconception that only endotherms can be surveyed using camera traps, I think it would be good to add ectotherms as a possible target group with specialised protocols (based on reference 44 in the manuscript).

Could the fact that the difference in functionality between camera traps and other methods was smaller in open habitats be an effect of other methods performing better rather than camera traps performing worse? I am thinking about line transects / distance sampling or aerial counts which work better in more open habitat and therefore one would expect a smaller difference between camera traps and other methods. Or is it just an effect of lack of trees and high + moving ground vegetation? [Line 317-325]

Figures

Figure 2: Maybe add a label for the last year that was taken into account in the meta-analysis (which is relatively hard to read now as one needs to count the bars since 2010).

Figure 4 and 5: It would be informative to add an asterisk or something to the comparisons where the 95% interval did not include 0.

Decision letter (RSOS-181748.R0)

25-Jan-2019

Dear Dr Wearn

On behalf of the Editors, I am pleased to inform you that your Manuscript RSOS-181748 entitled "Snap happy: camera traps are an effective sampling tool when compared to alternative methods" has been accepted for publication in Royal Society Open Science subject to minor revision in accordance with the referee suggestions. Please find the referees' comments at the end of this email.

The reviewers and handling editors have recommended publication, but also suggest some minor revisions to your manuscript. Therefore, I invite you to respond to the comments and revise your manuscript.

- Ethics statement

- Data accessibility

<http://datadryad.org/submit?journalID=RSOS&manu=RSOS-181748>

- Competing interests

- Authors' contributions

- Acknowledgements

- Funding statement

Please ensure you have prepared your revision in accordance with the guidance at <https://royalsociety.org/journals/authors/author-guidelines/> -- please note that we cannot publish your manuscript without the end statements. We have included a screenshot example of

the end statements for reference. If you feel that a given heading is not relevant to your paper, please nevertheless include the heading and explicitly state that it is not relevant to your work.

Because the schedule for publication is very tight, it is a condition of publication that you submit the revised version of your manuscript before 03-Feb-2019. Please note that the revision deadline will expire at 00.00am on this date. If you do not think you will be able to meet this date please let me know immediately.

Please note that Royal Society Open Science charge article processing charges for all new submissions that are accepted for publication. Charges will also apply to papers transferred to Royal Society Open Science from other Royal Society Publishing journals, as well as papers

submitted as part of our collaboration with the Royal Society of Chemistry (<http://rsos.royalsocietypublishing.org/chemistry>).

Kind regards,
Andrew Dunn
Senior Publishing Editor
Royal Society Open Science
openscience@royalsociety.org

on behalf of Prof Kevin Padian (Subject Editor)
openscience@royalsociety.org

Reviewer comments to Author:
Reviewer: 1

Comments to the Author(s)
See attached file.

Reviewer: 2

Comments to the Author(s)
I really enjoyed reading your manuscript entitled 'Snap happy: camera traps are an effective sampling tool compared to alternative methods'. I think it is a timely meta-analysis of the studies comparing camera trapping with other survey methods that confirms the big potential that is often attributed to camera traps. The methods for the meta-analysis are well described and appropriate statistics performed. I have only a few questions and minor suggestions for additions to the manuscript.

Material and methods

The authors made sure that only within study comparisons were made in order to reduce biases in the analyses due to differences in quantification of certain parameters (for example calculating detection probability per day or per week). This is further supported by the use of the log response ratio as the measure of effect size. If I am correct, this method assumes that parameters were calculated in the same way and on the same scale within studies. If the authors checked for this, it is good to make this explicit, otherwise, I think this needs checking to make sure that comparisons are made in a consistent way. [Lines 142-147]

Results

[Lines 223-225] You report a difference between factor levels, but which factor is taken as the baseline? I guess digital, but would be good to state.

For bait: where camera traps with bait compared to other methods with bait, and camera traps without bait with other methods without bait? Or could camera traps with bait be compared to other methods without bait and vice versa? I am wondering because of the counter intuitive results for baited cameras having a smaller difference with other surveys than un-baited cameras. Could this perhaps be because studies in which baited cameras were used were also more likely to use other methods that used bait, and the use of bait in general increased the effectiveness of any method, and therefore there is a smaller difference between methods? [Lines 228-234]

Discussion

[Lines 294-296] As it is a common misconception that only endotherms can be surveyed using camera traps, I think it would be good to add ectotherms as a possible target group with specialised protocols (based on reference 44 in the manuscript).

Could the fact that the difference in functionality between camera traps and other methods was smaller in open habitats be an effect of other methods performing better rather than camera traps performing worse? I am thinking about line transects / distance sampling or aerial counts which work better in more open habitat and therefore one would expect a smaller difference between camera traps and other methods. Or is it just an effect of lack of trees and high + moving ground vegetation? [Line 317-325]

Figures

Figure 2: Maybe add a label for the last year that was taken into account in the meta-analysis (which is relatively hard to read now as one needs to count the bars since 2010).

Figure 4 and 5: It would be informative to add an asterisk or something to the comparisons where the 95% interval did not include 0.

Author's Response to Decision Letter for (RSOS-181748.R0)

See Appendix B.

Decision letter (RSOS-181748.R1)

12-Feb-2019

Dear Dr Wearn,

I am pleased to inform you that your manuscript entitled "Snap happy: camera traps are an effective sampling tool when compared to alternative methods" is now accepted for publication in Royal Society Open Science.

You can expect to receive a proof of your article in the near future. Please contact the editorial office (openscience_proofs@royalsociety.org and openscience@royalsociety.org) to let us know if

you are likely to be away from e-mail contact. Due to rapid publication and an extremely tight schedule, if comments are not received, your paper may experience a delay in publication.

on behalf of Prof Kevin Padian (Subject Editor)
openscience@royalsociety.org

Appendix A

This was a well-written manuscript and a pleasure to read. With the increasing number of camera trap studies it is a timely contribution and provides important considerations for researchers planning wildlife surveys, including camera trap surveys. While I don't have major concerns over the general discussion or conclusions I do have some concerns with the statistical analysis, which I have outlined generally, and specifically, below.

Concerns with the Statistical Analysis

General comments:

The use of model-averaging requires some explanation/justification. Recent work highlights issues with model averaging and suggests that for inference one should use the full model (e.g., Bolker 2018) and use a priori model reduction (Harrell 2001). Please also mention if there was any collinearity among covariates (including VIF) and if covariates were standardised.

I am assuming that the authors are using the range of confidence intervals to indicate significant results -i.e., if confidence intervals overlap 0 then the effects are not significant. This is not explicitly stated in the methods. It would be useful if the authors were explicit on what they used to determine significance.

There were 32 and 128 models run, respectively. Considering these numbers we would expect, from chance alone, to have 5-6 significant results. Are the authors confident that the results are the truth rather than spurious?

Specific comments:

Line 197 and 198 - both statements require references.

Line 224 - the results are reported using model averaged estimates. There was no mention of model averaging in the methods.

Line 237 - the confidence intervals overlap 0, which I interpret to mean that the result is not significant - i.e., this meta-analysis **does not** show that camera traps have a higher effectiveness relative to other methods. Thus, I do not see how the results substantiate the claim made on Line 259 unless there is the removal of film camera traps (Line 308). Perhaps revise to include this result in the results section

and thereby provide the reader with all pertinent information before delving into the discussion.

Line 270-272 - can the authors expand on why these particular characteristics might have a more equivocal bearing on the effectiveness of camera traps? Is it that these characteristics are context dependent and/or that the sampling design of this meta-analysis was not able to disentangle the results?

Joanna M. Burgar
Postdoctoral Researcher,
University of Victoria; University of British Columbia

Appendix B

ZSL INSTITUTE OF ZOOLOGY

Dr Oliver R. Wearn
AXA Research Fellow
Institute of Zoology
Zoological Society of London
Regent's Park, London
NW1 4RY, UK
Tel: +44 (0)20 7449 6322
oliver.wearn@ioz.ac.uk
<https://www.zsl.org/science>

Prof Kevin Padian (Subject Editor)
Royal Society Open Science

25 February 2019

Dear Professor Padian,

We would like to submit our revised manuscript entitled "Snap happy: camera traps are an effective sampling tool when compared to alternative methods". We have addressed all of the very helpful comments made the reviewers and provide a point-by-point response attached to this letter.

We hope that our manuscript will be accepted for publication in *Royal Society Open Science* and very much look forward to hearing from you.

Yours sincerely,

Oliver Wearn and Paul Glover-Kapfer

Reviewer 1

This was a well-written manuscript and a pleasure to read. With the increasing number of camera trap studies it is a timely contribution and provides important considerations for researchers planning wildlife surveys, including camera trap surveys. While I don't have major concerns over the general discussion or conclusions I do have some concerns with the statistical analysis, which I have outlined generally, and specifically, below.

May we take this opportunity to thank you for reviewing the paper.

Concerns with the Statistical Analysis General comments:

The use of model-averaging requires some explanation/justification. Recent work highlights issues with model averaging and suggests that for inference one should use the full model (e.g., Bolker 2018) and use a priori model reduction (Harrell 2001).

We thank the reviewer for drawing our attention to the most recent statistical literature on model-averaging, which we must admit we were not aware of (the learning never stops...). We could not find any published work by Ben Bolker in 2018 or 2019 in this area, but found a draft manuscript on his GitHub page, which we assume the reviewer is referring to. This manuscript takes issue with the concept of model-averaging of parameter estimates, suggesting (on pg 3) instead to use the full model or conduct null-hypothesis significance testing. The former is undesirable when datasets are small, as is common in ecology (and also true in our case), because it can increase parameter uncertainty, potentially “to the point where we can't tell anything for sure” (Bolker manuscript, pg 3). The latter only yields p-values, not parameter estimates that also incorporate model uncertainty (as model-averaging does). This leaves us uncertain what approach exactly to follow.

We also note that, whilst Bolker's viewpoint might become generally-accepted practice in ecology in future, the reviewer will hopefully agree with us that there is considerable controversy and disagreement on these issues at present. In effect, Bolker appears to be arguing against the whole idea of model selection (i.e. just use the full model), which in our view is not in line with current consensus opinion. We have taken the (admittedly conservative) decision to retain our existing approach and now have provided readers with a signpost to the controversy surrounding model-averaging (with citations to two papers which are critical of model-averaging: Cade, 2015, *Ecology*, 96:2370-2382 and Banner & Higgs, *Ecol. Appl.*, 27:78-93). As ecologists, we evidently need some clarity on these issues and hope that the statistical community can deliver some in the short- or medium-term.

Finally, we checked what difference it would have made to our inference if we had used the full models for inference instead of model-averaging. We found that in practice the different approaches yield similar results, for our dataset at least.

Response variable	Parameter	Parameter estimate using model-averaging	Parameter estimate using full model
Author recommendations (proportional odds model)	Camera type	-1.4219*	-1.4480*
	Use of attractants	-0.9356*	-0.9893*
	Habitat openness	-0.4013	-0.3963
	Latitude	0.2035	0.1607
	Minimum weight	~	0.1190
Response ratios (mixed-effects model)	Metric - Precision of state variable	0.791657*	0.815443^
	Metric - Detection probability	0.263458	0.272817^
	Metric - Latency to detection	0.895426*	0.916852^

Metric - Detection rate	0.70945*	0.728415^
Metric - Species richness	0.422341	0.452581^
Method - Scat survey	0.35284	0.421605^
Method - Track plate	0.003277	-0.014615^
Method - Track plot	-0.023471	0.05459^
Method - Hair trap	1.232754*	1.216825^
Method - Live trap	0.426247	0.4626^
Method - Line transect	0.321768	0.450246^
Method - Plot survey	-0.039133	0.009067^
Camera type	-0.549267*	-0.592442*
Use of attractants	~	0.256503
Habitat openness	-0.408713*	-0.405325*
Latitude	~	-0.019534
Minimum weight	~	0.137759

~ Covariate not present in best model set; * Parameter deemed to be 'significant' either on the basis of 95% CIs (model-averaging) or likelihood ratio tests (full model); ^ The covariates *Metric* and *Method* as a whole were judged to be significant on the basis of likelihood ratio tests.

Please also mention if there was any collinearity among covariates (including VIF) and if covariates were standardised.

Our covariates were not collinear, as confirmed by variance inflation factors, as well as visual inspection of scatterplots. We have now added the following sentence: "We tested for collinearity among the covariates using variance inflation factors, which provided no cause for concern (all factors < 3; Zuur et al. 2010)".

We previously did not standardise the covariates because we did not include any interactions in models (and collinearity among covariates was minimal). However, further reading (prompted by the first comment, above) alerted us to recent recommendations to calculate standardised parameter estimates as complementary measures of variable importance (along with summed Akaike weights, as we had already calculated; Galipaud et al. 2017, *MEE* 8:1668-1678). We therefore re-analysed the data using standardized covariates. This made almost no difference to the results, both quantitatively and qualitatively, because the continuous covariates (Latitude and Minimum weight) were of limited importance anyway. We now state in our Methods that covariates were standardised, and report standardised parameter estimates in the text.

I am assuming that the authors are using the range of confidence intervals to indicate significant results -i.e., if confidence intervals overlap 0 then the effects are not significant. This is not explicitly stated in the methods. It would be useful if the authors were explicit on what they used to determine significance.

We had stated (lines 169-170) that we "judged effect sizes to be significant if the intervals did not overlap zero". We suspect that the reviewer may have inadvertently missed this.

For comparing two different effect size estimates, we had previously just checked if the two confidence intervals overlap. We now realise that we were overly conservative with this test (it corresponds to a test with an alpha closer to 0.005, rather than the intended 0.05). We now carry out post-hoc paired Wald tests. This has not affected any of our inferences, however (none of the tested differences are significant, as before).

There were 32 and 128 models run, respectively. Considering these numbers we would expect, from chance alone, to have 5-6 significant results. Are the authors confident that the results are the truth rather than spurious?

We agree with the reviewer that this is rather more models than is recommended. There is an obvious danger here of 'data mining'. However, unlike in the case where substantial theory can be drawn upon, we could not formulate specific hypotheses about which combinations of covariates would be more or less likely to be the 'true' model. We could only justify our decisions about which covariates we thought would be important, and we considered all models that we fit to the data to be equally plausible explanations for the data. This concurs with Grueber et al. (2011, *Evol. Biol.* 24:699-711): "[all-subsets modelling] is acceptable insofar as each model is ecologically justifiable". Our view also concurs with the pragmatic view of Symonds & Moussalli (2011, *Behav. Ecol. Sociobiol.* 65:13-21): "there often exists insufficient knowledge of the system under study such that explorative methods like the all-subset approach present above are the only way forward".

Although justifying our approach, none of this guarantees that our results are real. However, we note that, although we ran 160 models in total, we actually only carried out formal significance tests (i.e. null hypothesis significance testing) in checking whether 95% confidence intervals overlapped with zero, which we did 8 times (for the 4 variables that were in the top model set for the author recommendations, and for the 4 variables that were in the top model set for the response ratios). In addition, we did not focus on these results, and instead used the measures of relative variable importance (standardised parameter estimates and sums of Akaike weights) to determine the main drivers of variation in the author recommendations and response ratios.

Finally, we note that our inferences would have been identical had we just used the full models (e.g. as advocated by Ben Bolker), for example with null hypothesis significance testing, and not carried out model selection. To confirm this, we dropped each covariate from the full models in turn and calculated the 'significance' of the covariate using a likelihood ratio test. This process identified the same covariates with significant effects as the information criterion-based model selection and averaging (*Camera type* and *Use of attractants* for the author recommendations, and *Metric*, *Method*, *Camera type* and *Habitat openness* for the response ratios).

Specific comments:

Line 197 and 198 - both statements require references.

We have added suitable references for these statements.

Line 224 - the results are reported using model averaged estimates. There was no mention of model averaging in the methods.

We have fixed this omission and added a description of the model-averaging.

*Line 237 - the confidence intervals overlap 0, which I interpret to mean that the result is not significant - i.e., this meta-analysis **does not** show that camera traps have a higher effectiveness relative to other methods. Thus, I do not see how the results substantiate the claim made on Line 259 unless there is the removal of film camera traps (Line 308). Perhaps revise to include this result in the results section and thereby provide the reader with all pertinent information before delving into the discussion.*

We have moved the post-hoc result for digital camera trap alone to the Results, as suggested. This hopefully better prepares the reader for what follows in the Discussion.

Line 259 does not claim that camera traps have a significantly higher effectiveness relative to other methods, but instead just claims that our results "in general support the notion that

camera traps are a highly effective wildlife survey tool". We have therefore decided to leave this sentence as it is.

Line 270-272 - can the authors expand on why these particular characteristics might have a more equivocal bearing on the effectiveness of camera traps? Is it that these characteristics are context dependent and/or that the sampling design of this meta-analysis was not able to disentangle the results?

We have now re-worded this sentence to read: "Other characteristics of studies – the use of baits and lures, the body size of focal taxa and latitude – were not found to have a strong overall bearing on the effectiveness of camera traps, although this does not discount the fact they are likely to be important considerations in specific cases."

We thank the reviewer for querying our vague wording here. We believe that the unclear results with respect to these factors is due to both of the reasons given by the reviewer. However, we do not see this necessarily as a weakness of our meta-analysis approach. A meta-analysis is necessarily 'broad-brush' and focusses on identifying the most important drivers explaining variation in the data overall. There is no contradiction in finding that certain factors are not *overall* important in determining camera trap effectiveness, whilst acknowledging that they might be very important considerations for individual studies.

Joanna M. Burgar
Postdoctoral Researcher,
University of Victoria; University of British Columbia

Reviewer 2

I really enjoyed reading your manuscript entitled 'Snap happy: camera traps are an effective sampling tool compared to alternative methods'. I think it is a timely meta-analysis of the studies comparing camera trapping with other survey methods that confirms the big potential that is often attributed to camera traps. The methods for the meta-analysis are well described and appropriate statistics performed. I have only a few questions and minor suggestions for additions to the manuscript.

Our sincere thanks to you for taking the time to review the paper.

Material and methods

The authors made sure that only within study comparisons were made in order to reduce biases in the analyses due to differences in quantification of certain parameters (for example calculating detection probability per day or per week). This is further supported by the use of the log response ratio as the measure of effect size. If I am correct, this method assumes that parameters were calculated in the same way and on the same scale within studies. If the authors checked for this, it is good to make this explicit, otherwise, I think this needs checking to make sure that comparisons are made in a consistent way. [Lines 142-147]

We thoroughly checked that metrics were calculated consistently and also checked that they were on approximately the same scale, where we deemed this to be important for the metric concerned (e.g. we deemed detection rate to be scale-invariant). We state in the Methods that "most studies attempted to spatially and temporally match the deployment of each survey method (e.g. co-deploying methods at the same sampling points). Where this was obviously not the case, we limited comparisons to metrics that we considered would still be

valid". We also revisit this in the Discussion, acknowledging that poor spatial or temporal matching may have affected some of the studies and that this is a possible limitation of our meta-analysis.

Results

[Lines 223-225] You report a difference between factor levels, but which factor is taken as the baseline? I guess digital, but would be good to state.

We now state that our baseline was indeed *digital* camera traps.

For bait: where camera traps with bait compared to other methods with bait, and camera traps without bait with other methods without bait? Or could camera traps with bait be compared to other methods without bait and vice versa? I am wondering because of the counter intuitive results for baited cameras having a smaller difference with other surveys than un-baited cameras. Could this perhaps be because studies in which baited cameras were used were also more likely to use other methods that used bait, and the use of bait in general increased the effectiveness of any method, and therefore there is a smaller difference between methods? [Lines 228-234]

We were specifically interested in whether the use of an attractant *with camera traps* altered the effectiveness of camera-trapping, so cameras could have been paired with either baited or unbaited methods across studies. We have clarified this in the Methods.

The reviewer makes an interesting point that studies using attractants with camera traps are also likely to have used attractants with the other comparison survey methods or, more specifically, with *methods* that use attractants. We now include this in the Supplementary Discussion as a possible explanation for the result that attractants were apparently found to lower the effectiveness of camera traps according to the author recommendations.

Our models of the response ratios, however, accounted for this possible effect by virtue of the fact that the variation in the comparison survey method almost perfectly accounted for the variation in whether the comparison survey method used an attractant (e.g. live-trapping and track plates always used an attractant, whilst line transects and plot surveys never did). In this case, *Use of attractants* was not found to be important. We are therefore confident that attractants were not an important determinant of camera trap effectiveness as measured using response ratios.

Discussion

[Lines 294-296] As it is a common misconception that only endotherms can be surveyed using camera traps, I think it would be good to add ectotherms as a possible target group with specialised protocols (based on reference 44 in the manuscript).

We have made this change.

Could the fact that the difference in functionality between camera traps and other methods was smaller in open habitats be an effect of other methods performing better rather than camera traps performing worse? I am thinking about line transects / distance sampling or aerial counts which work better in more open habitat and therefore one would expect a smaller difference between camera traps and other methods. Or is it just an effect of lack of trees and high + moving ground vegetation? [Line 317-325]

We have now included in the Discussion this idea that other methods might be less effective in closed habitats (such as line-transects). We had previously just said that other methods might not be possible in closed habitats.

Figures

Figure 2: Maybe add a label for the last year that was taken into account in the meta-analysis (which is relatively hard to read now as one needs to count the bars since 2010).

We have made this change.

Figure 4 and 5: It would be informative to add an asterisk or something to the comparisons where the 95% interval did not include 0.

We have added asterisks to the 'significant' factors in figures 4 and 5, with a note in the captions to explain this.